# Safety, feasibility, acceptability and preliminary effects of the Neurofenix platform for Rehabilitation via HOMe Based gaming exercise for the Upper-limb post Stroke (RHOMBUS): results of a feasibility intervention study

Cherry Kilbride [1], Daniel J M Scott,[1,2] Tom Butcher,[1] Meriel Norris,[1] Alyson Warland,[1] Nana Anokye,[1] Elizabeth Cassidy,[3] Karen Baker,[1,2] Dimitrios A Athanasiou,[2] Guillem Singla-Buxarrais,[2] Alexander Nowicky,[1] Jennifer Ryan[1,4]

¹Department of Health Sciences, Brunel University London, London, UK
²Neurofenix, London, UK
³Freelance Academic and Research Supervisor, London, UK
⁴Public Health and Epidemiology, Royal College of Surgeons, Dublin, Ireland

**Correspondence to**
Dr Cherry Kilbride;
Cherry.Kilbride@brunel.ac.uk

## ABSTRACT

**Objectives** To investigate the safety, feasibility and acceptability of the Neurofenix platform for home-based rehabilitation of the upper limb (UL).

**Design** A non-randomised intervention design with a parallel process evaluation.

**Setting** Participants' homes, South-East England.

**Participants** Thirty adults (≥18 years), minimum 12-week poststroke, not receiving UL rehabilitation, scoring 9–25 on the Motricity Index (elbow and shoulder), with sufficient cognitive and communicative abilities to participate.

**Interventions** Participants were trained to use the platform, followed by 1 week of graded game-play exposure and 6-week training, aiming for a minimum 45 min, 5 days/week.

**Outcomes** Safety was determined by assessing pain and poststroke fatigue at 8 and 12 weeks, and adverse events (AEs). Impairment, activity and participation outcomes were measured. Intervention feasibility was determined by the amount of specialist training and support required to complete the intervention, time and days spent training, and number of UL movements performed. Acceptability was assessed by a satisfaction questionnaire and semistructured interviews.

**Results** Participants (14 women; mean (SD) age 60.0 (11.3) years) were a median of 4.9 years poststroke (minimum-maximum: 1–28 years). Twenty-seven participants completed the intervention. The odds of having shoulder pain were lower at 8 weeks (OR 0.45, 95% CI 0.24 to 0.83, p=0.010) and 12 weeks (OR 0.46, 95% CI 0.25 to 0.86, p=0.014) compared with baseline. Fugl-Meyer upper extremity, Motor Activity Log and passive range of movement improved. No other gains were recorded. Poststroke fatigue did not change. Thirty mild and short-term AEs and one serious (unrelated) AE were reported by 19 participants. Participants trained with the platform for a median of 17.4 hours over 7 weeks (minimum-maximum: 0.3–46.9 hours), equating to a median of 149 min per week. The median satisfaction score was 36 out of 40.

**Conclusion** The Neurofenix platform is a safe, feasible and well accepted way to support UL training for people at least three months poststroke.

**Trial registration number** ISRCTN60291412.

## Strengths and limitations of this study

► A feasibility trial developed following Medical Research Council Framework for the development of complex interventions.
► Mixed-method approach for answering feasibility and acceptability objectives.
► Fidelity activity data were objectively measured through the platform.
► Inclusive of people with higher levels of upper-limb impairment.
► No control group or blinded assessor.

## INTRODUCTION

Upper-limb (UL) dysfunction is a common and major contributor to physical disability following stroke.[1] Approximately two-thirds of stroke survivors experience UL impairment[2] and only 5%–20% achieve full UL recovery at 6 months poststroke.[3]

High intensity UL stroke rehabilitation improves UL recovery with a positive dose–response relationship for individuals in the acute and chronic phases after stroke.[4] UL training provided in rehabilitation settings typically falls below the hundreds of repetitions considered necessary to drive functional recovery.[5,6]

Virtual reality (VR) platforms using interactive computer-based games have emerged as a feasible and motivating way of supporting UL rehabilitation at intensities significantly higher than conventional therapy.[7 8] However, bespoke VR devices are often unwieldy, complex and expensive and while commercially available devices may be more accessible, they often demand sophisticated movement combinations which are not possible for many stroke survivors to perform.[9–11]

The Neurofenix platform is a novel, affordable and portable VR device specifically developed by stroke survivors, specialist neurophysiotherapists and bioengineers for gamification of poststroke UL rehabilitation (www.neurofenix.com). The platform enables and encourages stroke survivors to exercise their arm and hand at home with little or no assistance. The overall aim of this study was to investigate the safety, feasibility and acceptability of the platform for home-based rehabilitation of the UL. Additional objectives were (1) to assess the feasibility of conducting a definitive trial of the clinical and cost-effectiveness of the intervention, (2) to understand factors relating to people with stroke and the intervention that may impact on fidelity to the intervention and (3) to examine preliminary effects of the intervention.

## METHODS
The study is reported in accordance with guidance on reporting pilot and feasibility studies.[12] Further details on trial procedures are detailed in the protocol.[13]

### Patient and public involvement
Stroke survivors were involved in the iterative development of the Neurofenix platform. Two further stroke survivors acted in an advisory capacity during the study, providing input to the protocol and trial documentation, and dissemination.

Modifications to study design since trial registration
1. At the time of trial registration, the intervention was called Gameball device; it has since been renamed the Neurofenix Platform.
2. Measurement of pain—simplified to self-report using Visual Analogue Scale (VAS) 0–10 with a focus on shoulder pain.
3. Removal of EuroQoL5D5L overall health on the day VAS 0–100.
4. Addition of data analysis for preliminary effects of intervention (https://www.nihr.ac.uk/documents/nihr-research-for-patient-benefit-rfpb-programme-guidance-on-applying-for-feasibility-studies/20474).

### Trial design
The Rehabilitation via HOMe Based gaming exercise for the Upper-limb study used a non-randomised intervention design with a parallel process evaluation. Participants were familiarised, set-up and trained to use the device by a research therapist, followed by 1 week of graded exposure to game-play, and subsequent 6-week independent

training phase. The device was removed at the end of the 7-week training period. Assessments were performed at baseline, 8 and 12 weeks.

### Participants
Participants were recruited from a university stroke database, via gatekeepers for three community stroke groups and the ISRCTN Registry website between April and September 2018. Inclusion criteria were: (1) aged 18 years or over; (2) capacity to consent; (3) self-reported diagnosis of stroke; (4) 12 weeks minimum poststroke and not in receipt of UL rehabilitation; (5) mild-to-severe reduction in arm function poststroke (Motricity Index score between 9 and 25 for elbow and shoulder movement); (6) able to sit or stand independently, with or without an aid, for a minimum of 5 min and (7) sufficient English to participate in the intervention and assessments. Exclusion criteria were: (1) unstable medical conditions; (2) uncontrolled photosensitive epilepsy; (3) acquired brain injury from other causes, bilateral or cerebellar lesions; (4) uncompensated visual neglect, hemianopia or uncorrected visual field deficits and (5) pre-existing, unremitting arm pain at rest. All participants provided written informed consent.

### Intervention
The intervention is described in detail in the protocol.[13] Briefly, the Neurofenix platform is a digital device consisting of non-immersive VR software in the form of an App on a tablet and Bluetooth-connected hand controller (NeuroBall) and arm controllers (Neurobands). The NeuroBall is secured to the impaired hand through a system of straps and elastic finger holds. Motion sensors within the NeuroBall detect UL movements and translate these to control games displayed on a tablet. For those without sufficient strength and/or range of movement in the hand and wrist to control the NeuroBall, the Neuro-Bands are used as an alternative. The NeuroBands are two small motion sensors; the first straps around the upper arm and the second around either the forearm or the hand. If the second NeuroBand is placed on the forearm, then elbow flexion and extension can be used to control the games, and if placed around the hand, wrist flexion and extension can be used. Participants can choose from seven games specifically developed for UL rehabilitation (see www.neurofenix.com).

Two research therapists attended two half-day training sessions on technical and operational procedures relating to the device. Neurofenix engineers provided additional technical support to the therapists and participants throughout the study if required.

Following the baseline assessment, a research therapist attended the participant's home. The visit commenced with a brief questionnaire about the participant's prior experience with technology and video games and confidence in using new technology. The pace and depth of subsequent teaching was based on the participant's experience and confidence. The participant was then given

the Neurofenix platform consisting of a NeuroBall or two NeuroBands, a tablet, a tablet stand, chargers, a handbook and a one-page 'Quickstart' guide.

A research therapist trained the participant in how to use the platform independently or with the help of a carer if requested. Specifically, participants were taught safety precautions and were trained to (1) turn the device and tablet on and off, (2) don and doff the device, (3) calibrate the device, (4) play each of the games on the tablet, (5) navigate the menus, (6) track their progress, (7) use the handbook and (8) charge the devices. The research therapist advised each participant on the starting duration and frequency of use based on signs of pain and poststroke fatigue during the visit. As participants were not receiving UL rehabilitation at the time of the study, muscle ache as a normal response to training was discussed with the stroke survivor. Participants were advised to gradually increase the amount of use over the first week, with the aim of achieving a minimum of 45 min of training on 5 days a week, and to continue with this duration and frequency for the subsequent 6 weeks, increasing the dose if able. Participants independently chose which games to play during the intervention. Participants received two scheduled phone calls from the research therapists during week 1 and week 3 to ensure no clinical problems or technical faults had occurred. After this second call, participants had no input from the research team, but were advised to contact the team if they needed clinical or technical support. Participants were contacted at the end of 7 weeks, asked to stop using the device and arranged a time for the therapist to collect the device.

## Procedures

Two research therapists completed assessments in participants' homes. Socioeconomic characteristics, stroke-specific characteristics and global disability were measured using the simplified modified Rankin Scale questionnaire (smRSQ); higher score represents greater disability (range 1–5).[14 15]

## Outcomes

Arm and hand function were assessed objectively using the Action Research Arm Test (ARAT), higher scores indicate better function.[16–18] UL impairment was assessed using the Fugl-Meyer Assessment–Upper Extremity; higher scores indicate less impairment.[19 20] Participation was measured on the 10-item Subjective Index of Physical and Social Outcome (SIPSO); higher scores signify increased ability to reintegrate to a 'normal' lifestyle.[21] Self-reported arm use during functional tasks was measured using the 28-item Motor Activity Log (MAL); higher scores indicate more activity.[22 23]

The Modified Modified Ashworth Scale (MMAS) was used to assess spasticity of the shoulder adductors and internal rotators, elbow, wrist and finger flexors.[24] The highest spasticity score within the UL was selected for analysis. Passive range of movement (PROM) of the shoulder (flexion, abduction, external rotation), elbow (flexion and extension) and wrist (extension) were assessed using goniometry.[25] Quality of life was assessed using the EQ-5D-5L questionnaire.[26] A utility value (ie, a score) was calculated as recommended using the cross-walk function which maps the EQ-5D-3L and newer EQ-5D-5L questionnaires[27]; a higher value indicates better quality of life. A modified version of the Client Service Receipt Inventory (CSRI) was used to assess health service use.[28 29]

## Safety

Safety was determined by assessing pain, poststroke fatigue and adverse events (AEs). At 8 and 12 weeks, participants were asked if they experienced shoulder pain over the past 7 days, and if so to rate the intensity of shoulder pain on a 10-point VAS, with higher scores indicating more pain. Poststroke fatigue level was assessed using the Fatigue Severity Scale (FSS-7).[30] At the 8-week and 12-week assessment, participants were asked if they had experienced an AE since their last contact with the research team using standardised questioning. An AE is any untoward medical occurrence affecting a participant during the study which may or may not be related to the study intervention that is, an AE is unrelated if there is no evidence of a causal relationship with the trial intervention and another documented cause of the AE is most plausible. A serious AEs (SAEs) is an AE that results in death, is life threatening, requires hospitalisation or results in a disability or incapacity.

## Feasibility

Feasibility of the intervention was determined by recording the number of people who received the training sessions, the number and duration of sessions required, and the number and duration of clinical or technical phone calls and home visits required. Post-training confidence with the Neurofenix Platform was assessed with a 10-point VAS (higher score equates to more confidence).

Feasibility was also determined by assessing fidelity to the intervention. Fidelity was assessed as time spent training the UL with the platform, number of days training, and number of UL movements performed. Sensors were used to automatically collect data from the Neurofenix platform, which was stored on the participant's tablet until remotely downloaded by a member of the research team.

## Acceptability

The Quebec User Evaluation of Satisfaction with Assistive Technology (QUEST 2.0)[31] was completed at 8 weeks to assess user satisfaction of the device; higher scores indicate higher satisfaction. Semistructured interviews were completed with a purposive sample (male, female, high and low user engagement) of 18 participants to explore the acceptability of the intervention, and the experience of taking part in the study. Interviews were conducted by a research therapist (typically who did not provide training to that participant) and the analysis was undertaken by a member of the research team who did not otherwise

have contact with participants (EC). Carers or spouses involved in the intervention were also invited to take part as a dyad.

## Feasibility of a definitive trial

The feasibility of conducting a definitive trial was assessed by examining recruitment and retention rates, outcome measure completion and reasons for missing data associated with the data collection tools.

## Sample size

We included 30 participants based on recommendations that between 24 and 50 participants are sufficient for feasibility studies.[32 33]

## Data analysis

Distribution of data was explored using Q-Q plots, histograms and cross-tabulations. Descriptive statistics (eg, mean, SD, median, IQR) were used to report data. Generalised estimating equations with an exchangeable working matrix and robust standard errors were used to examine change between baseline and follow-up assessments on the VAS, FSS, ARAT, FMA-UE, MMAS, MAL-28 (amount of use score only), SIPSO and goniometry. A generalised estimating equation with a logit link, exchangeable working matrix and robust standard errors was used to examine if the odds of shoulder pain changed over time.

Associations between total time spent exercising the UL with the platform, number of repetitions performed and age, ARAT score at baseline, FMA-UE score at baseline, confidence with new technology at baseline, post-training confidence with the Neurofenix platform were examined using Spearman's correlations. Associations between total time spent exercising the UL with the platform, number of repetitions performed, and gender were examined using Mann-Whitney U tests.

Qualitative data from the semistructured interviews were analysed using the five stage Framework Analysis method.[34] Framework Analysis incorporates both deductive and inductive coding and enabled trial processes and experiences to be explored and reported. This method provides a strong audit trail of the analytical process, which enhances transparency.[35] One person (EC) independent to the delivery team led the analysis. Four researchers (EC, MN, TB and DJMS) independently coded the same three transcripts, then discussed and agreed codes. A coding framework was then developed and used to code all the interviews.

## RESULTS

Thirty participants were recruited to the study between April and July 2018 (figure 1). Participants (14 women) were mean age (SD) 60.0 (11.3) years (minimum-maximum: 36–85 years), and a median of 4.9 years poststroke (minimum-maximum: 1–28 years) (table 1). The majority scored 2 or 3 on the smRSQ. Mean (SD)

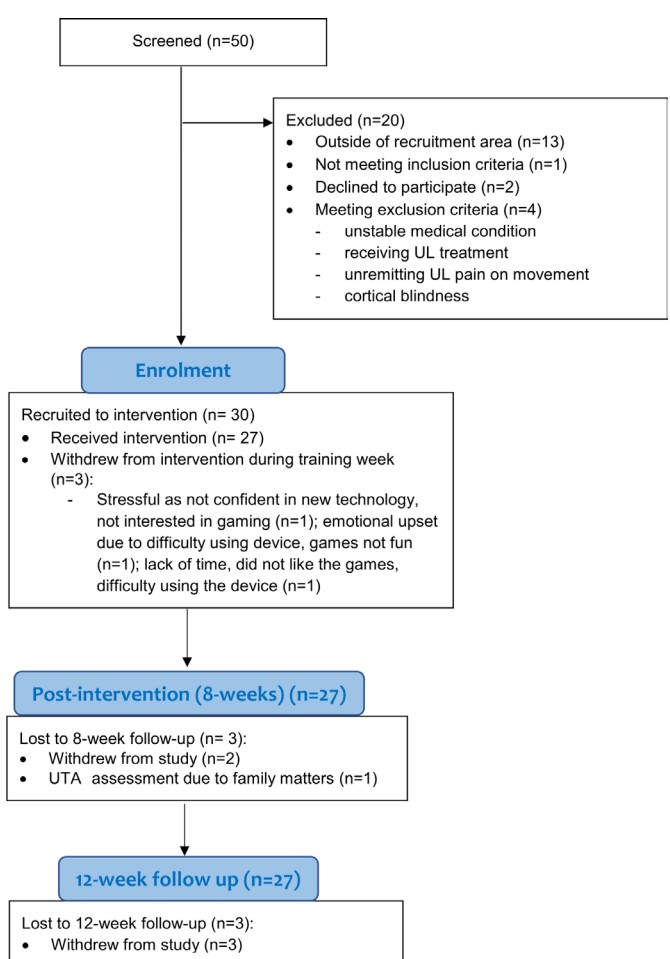

**Figure 1** CONSORT flow of participants through the trial. CONSORT, Consolidated Standards of Reporting Trials; UL, upper limb; UTA, Unable to Attend.

FMA-UE score was 33 (16.4) and median (IQR) ARAT score was 8 (20).

## Safety

The odds of having shoulder pain were lower at 8 weeks (OR 0.45, 95% CI 0.24 to 0.83, p=0.010) and 12 weeks (OR 0.46, 95% CI 0.25 to 0.86, p=0.014) compared with baseline. Mean intensity of shoulder pain, as assessed on a VAS, was lower at 12 weeks compared with baseline (mean difference −0.69, 95% CI −1.37 to 0.004, p=0.052; table 2). There was no change in the level of poststroke fatigue as measured by the FSS-7 score, at 8 or 12 weeks in comparison to baseline (table 2). Thirty AEs and 1 SAE were reported by a total of 19 participants (table 3). Nineteen AEs were deemed probably or possibly related to the intervention; the AEs were mild and short-term and no participants withdrew due to an AE. Eleven unrelated AEs (including four falls) were deemed to have no causal relationship with the trial procedures as determined by the principal investigator (CK) and in line with the principles of Good Clinical Practice (https://www.hra.nhs.uk/planning-and-improving-research/policies-standards-legislation/good-clinical-practice/).

**Table 1** Baseline characteristics of RHOMBUS participants

|  | n (%) | Mean (SD) | Median (IQR), minimum-maximum |
|---|---|---|---|
| Age, year | 30 | 60 (11.3) | 60 (12), 36–85 |
| Women | 14 (47) | – | – |
| Ethnicity |  |  |  |
| White | 15 (50) | – | – |
| Asian | 12 (40) | – | – |
| Black | 2 (7) | – | – |
| Mixed | 1 (3) |  |  |
| Stroke chronicity, year | 30 | 5.5 (5.4) | 4.9 (4.5), 1–28 |
| Stroke type (self-report) |  |  |  |
| Haemorrhagic | 17 (57) |  |  |
| Ischaemic | 12 (40) |  |  |
| Not known | 1 (3) |  |  |
| smRSQ |  |  |  |
| 1.0 | 3 (10) |  |  |
| 2.0 | 11 (36) |  |  |
| 3.0 | 13 (43) |  |  |
| 4.0 | 1 (3) |  |  |
| 5.0 | 2 (6) |  |  |
| FMA-UE (0–66) | 30 | 33 (16.4) | 33 (31), 8–63 |
| ARAT (0–57) | 30 | 16 (17.5) | 8 (20), 0–57 |
| MMAS worse score (0–5) | 30 | 2 (0.98) | 2 (2), 0–3 |
| NIHSS | 30 | 4.1 (3.1) | 3 (3), 1 –14 |
| Played computer games before stroke (Yes) | 19 (63) | – | – |
| Played computer games since stroke (Yes) | 15 (50) | – | – |
| Owned a tablet, computer, console or smartphone (Yes) | 28 (93) | – | – |
| Confidence pre training with new technology (0–10) | 30 | 5.63 (2.7) | 5 (4) |
| Confidence with NeuroPlatform post training (0–10) | 29* | 7.6 (2.24) | 8 (3) |

*One participant withdrew during training.
ARAT, Action Research Arm Test; FMA-UE, Fugl Meyer Assessment Upper Extremity; MMAS, Modified Modified Ashworth Scale; NIHSS, National Institute of Health Stroke Scale; RHOMBUS, Rehabilitation via HOMe Based gaming exercise for the Upper-limb post Stroke; smRSq, simplified modified Rankin Scale Questionnaire.

## Feasibility

Thirty participants completed the training session (table 4). Twenty-five participants needed one training session, three needed two sessions and two needed a third session. The median time for training, over 1–3 sessions, was 98 min per participant (minimum-maximum 70–290 min). Median (IQR) participant confidence with the device after the initial training session was 8 (2) out of 10. Nine participants (30%) contacted the research team for technical support during the intervention period. Five

**Table 2** Change in scores on 7-Item Fatigue Severity Scale (FFS-7) and Visual Analogue Scale for pain (VAS) between baseline and 8, and baseline and 12 weeks

|  | Baseline (n=30) | 8 weeks (n=27) | Change; 8 weeks – baseline (n=27) | 12 weeks (n=27) | Change; 12 weeks – baseline (n=27) |
|---|---|---|---|---|---|
|  | Mean (SD) | Mean (SD) | Mean difference (95% CI); p value | Mean (SD) | Mean difference (95% CI); p value |
| Fatigue FSS-7* (7-49) | 28.5 (13.2) | 25.8 (13.9) | −1.59 (−6.51 to 3.34); 0.528 | 26.1 (12.5) | −1.98 (−5.75 to 1.79); 0.304 |
| VAS pain (0–10) | 2 (2.24) | 1.7 (2.10) | −0.10 (−0.73 to 0.53); 0.756 | 1.22 (1.60) | −0.69 (−1.37 to 0.004); 0.052 |

*n=26 for 8 weeks.

**Table 3** Adverse events (AEs) and serious AEs (SAEs) reported during study

| Event type | No of events | Attribution | | | People reporting event, (n) |
| | | No of probably related events | No of possibly related events | No of unrelated events | |
|---|---|---|---|---|---|
| SAE | 1 | 0 | 0 | 1 | 1 |
| AE | 30 | 17* | 2† | 11‡ | 19 |

*Muscle fatigue/soreness/discomfort (n=9); eye strain (n=2); upper-limb pain (n=4), self-reported increase in spasticity (n=2).
†Neck pain (n=2).
‡Fatigue (n=2); falls (n=4); blurred vision (n=1); headache (n=4).

participants (17%) contacted the research team for clinical support. The total number of visits, calls and their median durations are reported in table 5.

### Fidelity to the intervention

Participants trained with the Neurofenix platform for a median (minimum-maximum) of 17.4 (0.3–46.9) hours over 7 weeks, equating to a median of 149 min per week. Participants trained for a mean (SD) of 25.4 (15.9) days over 7 weeks (minimum-maximum: 1–48 days), equating to mean (SD) 3.6 (2.3) days per week. Eight participants (26.7%) achieved at least 225 min per week during the study, which equates to the recommended 45 min of training per day on 5 days per week. Participants performed a median (minimum-maximum) of 15 092 (222–43 999) UL movements over 7 weeks.

No association was found between time spent training over 7 weeks and age (r=0.24, p=0.203), ARAT (r=0.06, p=0.744), FMA-UE scores (r=0.003, p=0.987), confidence with new technology (r=0.13, p=0.505), confidence post-training (r=−0.05, p=0.784). There was also no difference in time spent training between men and women (p=0.061). No association was found between number of UL movements over 7 weeks and age (r=0.18, p=0.351), ARAT (r=0.10, p=0.614), FMA-UL (r=0.06, p=0.761), confidence with new technology (r=0.15, p=0.427), confidence post-training (r=−0.005, p=0.979). There was also no difference in number of UL movements between men and women (p=0.105).

### Acceptability

The median score on the QUEST was 36 out of 40 (range 21–40). The device was well received and provided motivating UL training (table 6). Motivating factors included enjoyable games, positive in-game feedback, competition and challenge, engagement with more purposeful games

than prescribed home exercises. Participants suggested that a greater variety of games and more challenging levels might enhance engagement with the platform. The home-based self-directed training provided by the device and its compact size was positively appraised because it offered a structured practice schedule and set amount of time to train and enabled practice to be planned around daily life.

### Preliminary effects

The mean difference in outcomes between baseline and 8 and 12 weeks, respectively, is reported in table 7. Scores on the FMA-UE (mean difference 2.68, 95% CI 0.86 to 4.49; p=0.004) and MAL (mean difference 0.19, 95% CI 0.06 to 0.32; p=0.004) improved between baseline and 8 weeks. SIPSO total score increased between baseline and 12 weeks (mean difference 2.17, 95% CI 0.32 to 4.01; p=0.021). Passive range of shoulder external rotation, elbow flexion and wrist extension increased between baseline and 8 weeks, and between baseline and 12 weeks (p<0.05; table 7). No changes were found for other outcomes.

### Feasibility of conducting a definitive trial

Fifty people expressed interest in participating in the study. Twenty were excluded for reasons outlined in figure 1, resulting in 60% being recruited. Three people withdrew from the intervention; reasons for withdrawing are shown in figure 1. All participants completed all outcome measures at baseline. Of the three people who withdrew from the intervention, two withdrew from the study and did not complete assessments at 8 or 12 weeks. One additional person did not complete the 8-week and 12-week assessment, respectively, resulting in 90% retention at follow-up assessments. In addition to the 10% missing data due to lost to follow-up, the FSS-7 was not

**Table 4** Training sessions required per participants and duration of training

| | No of participants receiving, n (%) | Duration, minutes | | |
| | | Mean (SD) | Median (IQR) | Minimum-maximum |
|---|---|---|---|---|
| Training sessions | | | | |
| First training session | 30 (100) | 95.23 (19.5) | 92 (25) | 55–150 |
| Second training session | 3 (10) | 48.33 (35.5) | 55 (70) | 10–80 |
| Third training session | 2 (6.7) | 95.0 (7.1) | 95 (10) | 90–100 |

**Table 5** Total number of phone calls and home visits (technical and clinical)

| | No of calls/visits | Duration, minutes per call/visit | | |
| --- | --- | --- | --- | --- |
| | | Mean (SD) | Median (IQR) | Minimum-maximum |
| **Clinical** | | | | |
| Phone calls | 10 | 12.0 (6.4) | 10 (9) | 4–19 |
| Home visits | 5 | 24.0 (8.2) | 30 (15) | 15–30 |
| **Technical** | | | | |
| Phone calls | 17 | 13.0 (10.3) | 13 (10) | 2–36 |
| Home visits | 15 | 49.0 (41.3) | 35 (62.5) | 5–123 |

assessed at 8 weeks, and shoulder external rotation was not assessed for one person at 12 weeks due to human error, and one person refused to complete the ARAT at 12 weeks. Data about medication dosage in the modified CSRI was missing for four participants.

## DISCUSSION

The results of this feasibility study indicate that the Neurofenix is a safe, feasible and well accepted intervention for UL rehabilitation for participants more than 1-year poststroke with mild-to-severe arm impairment. Participants experienced a reduction in shoulder pain over time, no

change in poststroke fatigue, and mild short lived AEs. Participants were highly satisfied with the rehabilitation programme (see table 6). Additionally, lost to follow-up and missing data were minimal, which in combination with data on safety, feasibility and acceptability, suggest that a definitive RCT is feasible.

On average, participants trained with the Neurofenix platform 4 days per week, for 149 min per week. Twenty-seven per cent achieved at least 225 min training per week which equates to the recommended 45 min of training per day for 5 days per week.[36] It may seem surprising, given the theoretical appeal of gamified therapy and

**Table 6** Illustrative quotations from the qualitative interviews

| Participant no and pseudonym | Quotation (page and line numbers) |
| --- | --- |
| **Training and support** | |
| P16 Ann (moderately impaired upper limb) | 'He went through everything. He went through all the games as well, which was good, so that helped. So, it just gives you confidence and reassurance in what you're doing.' (7, 266-268). |
| P22 Bal (moderately impaired upper limb): | 'If I didn't understand or if the response wasn't what I was expecting then I would refer to the detailed instructions to see if I could improve (5, 192-94). Very easy [to use] (…)The instructions were very clear (6, 229-237).(…)it's always useful to fall back on if you, if you can't cope.' (6, 247-48). |
| P19 Sam (moderately impaired upper limb): | 'Like I was stuck on a game, so I used the quick start just to find out if I was doing something wrong.' (13, 553-54). |
| P28 Terry's wife Fran | 'I thought the quick start guide on that sheet was very helpful 'cos I'd often look at that while we were setting you up, weren't… didn't I, Terry, and that was very helpful, but then we didn't need it as time went on.' |
| **Acceptability** | |
| P1 Iris (moderately impaired upper limb) | ' I was trying to get to level 20(…)Yeah, I was getting into it(…)I enjoyed the game because every time… If you didn't get to a, certain point it will always say, 'Re-try' or 'Re-play' or 'Start again', and I like that (17, 777-94). And the message after was, was to tell you how many minutes you played for and 'Congratulations, see you tomorrow, bye', and that's it, I like that.(…)It was like you had a friend talking to you.' (21, 979-80). |
| P6 Mark (mildly impaired upper limb): | 'It was a little hard at first, right, because I used to be pressing it and pressing it but, er, anyth… anyway I got used to it like.' (21, 956-57). |
| P16 Ann (moderately impaired upper limb) | 'I like the ones that I can, that I actually can achieve, yeah. It's no good…. I don't like ones that I know that I'm never gonna get to or never gonna do, I like ones that I can… Yeah, the easier ones for me that I can achieve relatively easy. Obviously it's gotta push you a little bit but not too much.' (14, 648-51) |
| P22 Bal (moderately impaired upper limb) | 'I have found it, er, quite entertaining but sometimes, because, er, there's not much variety, it's the same music, the same games, it's a bit boring to do it every day.(…)If there was a bit more variety or slightly different games, or even the games you could adjust so that, er, it was, er, slightly different, then I think it might not be so boring.' (15, 650-58). |

**Table 7** Change in outcomes between baseline and 8 weeks and baseline and 12 weeks

| Outcome | Baseline (n=30) Mean (SD) | 8 weeks (n=27) Mean (SD) | Change; 8 weeks−baseline (n=27) Mean difference (95% CI); p value | 12 weeks (n=27) Mean (SD) | Change; 12 weeks−baseline (n=27) Mean difference (95% CI); p value |
|---|---|---|---|---|---|
| ARAT* (0–57) | 15.9 (17.5) | 16.2 (17.3) | 0.17 (−0.86 to 1.21); 0.743 | 18.2 (19.1) | 0.63 (−0.97 to 2.23); 0.440 |
| FMA-UE (0–66) | 32.6 (16.4) | 35.2 (17.6) | 2.68 (0.86 to 4.49); 0.004 | 34.4 (17.8) | 1.49 (−0.38 to 3.37); 0.118 |
| SIPSO; total score (0–40) | 21.9 (9.3) | 23.5 (9.3) | 1.18 (−0.81 to 3.18); 0.245 | 24.3 (8.1) | 2.17 (0.32 to 4.01); 0.021 |
| MAL (0–5) | 0.89 (1.19) | 1.08 (1.16) | 0.19 (0.06 to 0.32); 0.004 | 1.10 (1.19) | 0.13 (−0.02 to 0.28); 0.080 |
| MMAS (0–5) | 2.0 (1.0) | 1.8 (1.0) | −0.16 (−0.38 to 0.06); 0.157 | 2.1 (1.1) | 0.08 (−0.19 to 0.35); 0.560 |
| EuroQol-5D-5L; utility score | 0.49 (0.32) | 0.56 (0.30) | 0.07 (−0.009 to 0.142); 0.088 | 0.54 (0.32) | 0.44 (−0.02 to 0.10); 0.145 |
| Shoulder flexion† | 121.3 (18.2) | 123.8 (19.4) | 3.09 (−2.17 to 8.37); 0.249 | 119.9 (21.2) | −0.26 (−7.31 to 6.77); 0.940 |
| Shoulder external rotation*† | 34.0 (19.9) | 41.5 (16.7) | 7.10 (2.36 to 11.84); 0.003 | 40.6 (16.4) | 6.52 (0.02 to 13.03); 0.049 |
| Shoulder abduction† | 107.6 (17.0) | 111.1 (19.3) | 3.21 (−2.00 to 8.42); 0.228 | 108.7 (19.7) | 1.48 (−4.66 to 7.63); 0.636 |
| Elbow flexion† | 142.3 (8.7) | 145.9 (5.6) | 3.28 (0.44 to 6.12); 0.024 | 148.4 (6.3) | 6.00 (3.20 to 8.80); <0.001 |
| Elbow extension† | −4.26 (10.84) | −4.14 (12.06) | 0.54 (−1.94 to 3.03); 0.669 | −4.29 (11.00) | 0.39 (−1.79 to 2.58); 0.724 |
| Wrist extension† | 58.5 (29.1) | 67.1 (19.6) | 10.18 (1.31 to 19.05); 0.024 | 67.81 (23.22) | 10.9 (1.8 to 20.1); 0.019 |

*n=26 at 12 weeks.
†Passive range of movement, degrees.
ARAT, Action Research Arm Test; FMA-UE, Fugl-Meyer Assessment–upper extremity; MAL, Motor Activity Log; MMAS, Modified Modified Ashworth Scale; SIPSO, Subjective Index of Physical and Social Outcome.

its potential advantages over traditional home exercise programmes, that a larger proportion of participants were unable to achieve the recommended minimal dose. However, as community-based rehabilitation opportunities for stroke survivors typically reduce over time and rarely continue beyond 6 months[37] devices such as the Neurofenix platform offer a tangible means of increasing UL training hours from what is likely to be a very low base.[38] We do not have community-based therapy comparison figures for other forms of UL therapy, but in the acute setting, where arguably patients have more access to therapists and UL treatment, only 7.9 min a day of combined occupational therapy and physiotherapy for the UL during the first few weeks post stroke has been reported.[39] Our participants also achieved a median (minimum-maximum) of 15 092 (222–43 999) UL movements over 7 weeks, which compares favourably to an average of 32 movement repetitions per session of inpatient UL rehabilitation reported by Lang et al.[40] So while adherence is variable in this study, problems related to persistence with technology-enhanced UL rehabilitation are consistent with previous studies using rehabilitation technology.[41 42] A recent systematic review concluded that further research is necessary to understand factors underlying perseverance with home-based technology-enhanced UL training poststroke.[43] Standen et al[42] reported that family support, where available, was crucial to successful game-play. We also found that participants with severe UL impairment were dependent on carers for setup. Participants starting from a low baseline may apply an equivalent effort as those with less impairment without meeting training targets which suggests that personalised training goals may also be useful.

Our findings on safety are consistent with similar studies.[41 43 44] Self-limited use of the device permitted self-management of the level of training related muscle fatigue and may explain the improvement in pain. We systematically assessed AEs, regardless of their relatedness to the intervention, which may explain the relatively high proportion of participants who experienced at least one event. However, AEs were mild, and the majority were unrelated to the intervention, as found in comparable studies.[41 43]

Recruitment targets were achieved within the specified time frame of 5 months; 60% of those screened were enrolled in the trial. The eligibility to enrolment ratio compares favourably to a multicentre study that examined the use of the Nintendo Wii for UL stroke rehabilitation in the UK where a low enrolment figure (4%) was achieved using broad eligibility screening in ten stroke centres.[41] Our strategy of using three recruitment streams, targeting local stroke groups and newsletters for stroke survivors, positively influenced engagement with the study and enrolment. However, other modes of recruitment may be necessary to enrol sufficient numbers for a definitive trial. Attrition was within acceptable limits (<20%),[45] and, compared with similar trials, consistent with figures reported by Adie et al[41] and lower than those reported by Standen et al.[42] Both studies provided comparable levels of support and participants were of similar age to those in our study. There are no obvious differences to account for the increased drop out in the Standen et al trial.[42]

Outcome data reporting was very good. Some participants found the assessments time consuming and fatiguing. We assessed the four core measures recommended for stroke trials.[46] We also included safety measures for pain and poststroke fatigue, and the SIPSO because there is no consensus about the most appropriate measure of participation in stroke trials.[46] To balance the

need to fully test our research questions against the need to use outcome measures judiciously,[46] we additionally measured spasticity, PROM and the MAL.

Most participants in this trial required one training session to confidently use the device and no further in-trial technical or clinical support. These results compare favourably with Standen et al[42] suggesting the training and follow-up support required to implement this intervention is feasible in clinical practice. A minority of participants required phone calls and home visits to help with clinical and technical issues. These problems may impede fidelity and, as previously reported,[42] are potential barriers to independent home use of rehabilitation devices. The Neurofenix platform was designed to enable independent home-based rehabilitation. This study showed that although technology was effectively leveraged to reduce clinical contact time, individually paced home-based instruction and support services were necessary to effectively deliver the intervention. These additional hours would need to be factored into a fully scaled trial to ensure that technological interventions are not studied as a dose matched control but on how they would be rolled out and used by service users and clinicians in real life. A future RCT should examine the cost-effectiveness of the intervention compared with usual care, given the minimal therapist time involved in providing this intervention in a community setting.

Consistent with our findings, previous studies reported that participants generally enjoy using VR rehabilitation devices.[43 47 48] The compact size and light weight of the devices and the tablet facilitated the use of the platform in the home environment and positively influenced satisfaction and acceptability. Qualitative data suggested that a broader selection of games with a wide range of difficulty levels and the introduction of real time in-play competition and comparison may increase motivation and encourage persistence. These findings are consistent with game theory for stroke rehabilitation which emphasises the importance of in-built skill level adaptation, feedback, competition and socialisation.[49] Adaptations to the games may be required prior to conducting a definitive trial to increase motivation and persistence.

To be maximally inclusive, stroke survivors with moderate to severe arm impairment were included in the study which may be considered a strength. Participants were relatively young compared with the UK average age of onset for stroke[50] but with a comparable age distribution to participants included in similar studies.[42 43 51] Older people may be thought to be less inclined to use rehabilitation technology, but we found no associations between age and time spent training using the device. The Neurofenix platform was custom designed for therapeutic use, which confers advantages over commercial gaming devices.[49] The device calibrates individual movement repetitions but key design features that further augment motivation and persistence such as personalisation through in-game adaptation[49 52] were not included in our device at the time of the study.

## Study limitations

There are methodological limitations to this study. The study design did not include a control group, while in keeping with a feasibility study this did not allow us to test the willingness of participants to be randomised in a definitive RCT. Furthermore, the study used non-blinded assessors which may have generated higher effect estimates than blinded assessors.[53] Participants were relatively young compared with the UK average age of onset for stroke[50] but with a comparable age distribution to participants included in similar studies.[42 43 51] Stroke survivors can experience pain from varied stroke related causes such as central poststroke pain, pain from spasticity or contractures as well as from pre-existing comorbidities; in addition some stroke survivors also had communication difficulties which made the measurement of pain challenging.[54]

## CONCLUSION

The Neurofenix platform is a safe, feasible and well-accepted means of delivering UL rehabilitation to people following stroke. Iterative design continues to develop and improve the platform by incorporating game theory to encourage motivation and persistence. As the effects of stroke vary considerably across individuals, and as individuals are likely to have different technical capacities, a supported remote training model such as that used in this trial is likely to be needed in a definitive trial.

**Acknowledgements** The authors would like to thank two stroke survivors who assisted the development of the intervention and advised on the protocol, trial documentation and dissemination. Further thanks to the group facilitators of Different Strokes and the Action for Rehabilitation from Neurological Injury. Thanks to Professor Christina Victor for her support.

**Contributors** CK and GS-B conceived the study. DJMS, TB, AN, JR, NA, AW and MN designed the study. GS-B, DAA, DJMS, TB and CK designed the intervention. CK led the running of the trial. DJMS and TB led the collection, management and analysis of the data. KB led blinded ARAT assessment. MN led the process evaluation. JR led the statistical analysis. NA led the evaluation of the feasibility of the economic evaluation. EC led the qualitative analysis. All authors read and approved the final manuscript. CK is the study guarantor.

**Funding** This work was supported by Innovate UK grant number 104188.

**Disclaimer** The views expressed are those of the authors and not necessarily those of Innovate UK.

**Competing interests** GS-B and DAA are employed by Neurofenix (United Kingdom), a digital therapeutics company developing a therapy platform and sensor-based devices to augment rehabilitation. Neurofenix provided the Neurofenix platforms and technical support to the research therapists. Neurofenix had no influence on the design of the study, data collection, analysis and interpretation of the data or manuscript preparation. KB and DJMS were employed by Brunel University London during the period of the study, and are now Neurofenix employees.

**Patient and public involvement** Patients and/or the public were involved in the design, or conduct, or reporting, or dissemination plans of this research. Refer to the Methods section for further details.

**Patient consent for publication** Not applicable.

**Ethics approval** This study involves human participants and was approved by Brunel University London 10249-MHR-Mar/2018-12322-2. Participants gave informed consent to participate in the study before taking part.

**Provenance and peer review** Not commissioned; externally peer reviewed.

**Data availability statement** No data are available. Participants did not consent for datasets to be stored or accessed outside of the research team. Therefore, no datasets have been made publicly available.

**ORCID iD**
Cherry Kilbride http://orcid.org/0000-0002-2045-1883

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
