## [Reviewer comments · BMJ Open]

ARTICLE DETAILS

TITLE (PROVISIONAL)	Safety, feasibility, acceptability, and preliminary effects of the Neurofenix Platform for Rehabilitation via HOME Based gaming exercise for the Upper-limb post-stroke (RHOMBUS): results of a feasibility intervention study.
AUTHORS	Kilbride, Cherry; Scott, Daniel J M; Butcher, Tom; Norris, Meriel; Warland, Alyson; Anokye, Nana; Cassidy, Elizabeth; Baker, Karen; Athanasiou, Dimitrios; Singla-Buxarrais, Guillem; Nowicky, Alexander; Ryan, Jennifer

VERSION 1 – REVIEW

REVIEWER	Garcia-Agundez, Augusto Technische Universitat Darmstadt, Multimedia Communications Lab
REVIEW RETURNED	10-Jun-2021

GENERAL COMMENTS	In the abstract conclusion (page 4, line 5) and in the conclusion (page 23, line 29) I think using the term "acceptable" is misleading, as acceptable would rather indicate its viability as a rehabilitation procedure, which is linked to its effect more than to its acceptance. "well-accepted" could be a suitable replacement. In the sample size section, authors state "We estimated 30 individuals would be sufficient". How was this estimated? Concerning AEs, why do authors claim there were 11 unrelated AEs? What is the specific criteria to determine a certain AE as related or unrelated? Finally, and although not related to the study design itself, I am interested in the lack of adherence. Authors state that, ideally, patients should perform their rehabilitation for approximately 45 minutes per day and 5 times per week. I understand about 25% of participants did exercise this much. Is there any available information on what is the adherence to a comparable therapy plan (that is, 225 minutes per week) using different (e.g. non-gamified) approaches? Given that the theoretical advantage of gamified therapy is that it should be more appealing and thus increase adherence, this is worrying. This should also perhaps be described in the discussion in more detail, comparing it with related approaches and potential solutions.
---

REVIEWER	Dorsch, Simone South Western Sydney Local Health District, Bankstown-Lidcombe Hospital
REVIEW RETURNED	23-Jun-2021

GENERAL COMMENTS	This is a very clearly written manuscript and I commend the authors for their clear language and easy layout of information. I have minor suggestions only. Abstract Results do not contain the UL impairment/ activity outcomes - it would be good to read these in the abstract Methods Outcomes - it would be good to explain what is meant by 'a utility value using the cross-walk function' as this is not as familiar as other outcome measures Safety - in this description, it is not clear if pain and fatigue are adverse events. If fatigue is being called an adverse event it would be good to understand if the intervention was related to increased fatigue and the severity of the fatigue. It is also arguable as to whether fatigue in the context of exercise is an adverse event, as exercise that does not induce some fatigue is probably not intensive enough to be of benefit.
--

VERSION 1 – AUTHOR RESPONSE

Reviewer: 1

1. In the abstract conclusion (page 4, line 5) and in the conclusion (page 23, line 29) I think using the term "acceptable" is misleading, as acceptable would rather indicate its viability as a rehabilitation procedure, which is linked to its effect more than to its acceptance. "well-accepted" could be a suitable replacement.

Thank you for this suggestion, we have adjusted the term 'acceptable' to 'well-accepted' throughout the manuscript

2. In the sample size section, authors state "We estimated 30 individuals would be sufficient". How was this estimated?

We acknowledge the use of the word 'estimated' is misleading and have re-written the sentence as follows: We included 30 participants based on recommendations that between 24 and 50 participants are sufficient for feasibility studies.[32,33]

3. Concerning AEs, why do authors claim there were 11 unrelated AEs? What is the specific criteria to determine a certain AE as related or unrelated?

In keeping with the International Conference of Harmonisation and the World Health Organisation's principles of Good Clinical Practice each AE (or SAE) is characterised as causally related to the intervention or not. It is usual for the site trial team to determine who has the final say as to whether it is unrelated/ possibly related /probably related; as was the case in this study this usually falls to the Principle Investigator. The PI will consider such criteria as the temporal relationship, the participant's underlying clinical condition, the biologic/theoretical plausibility of relationship and a process of de-challenge and re-challenge.

4. Finally, and although not related to the study design itself, I am interested in the lack of adherence. Authors state that, ideally, patients should perform their rehabilitation for approximately 45 minutes

per day and 5 times per week. I understand about 25% of participants did exercise this much. Is there any available information on what is the adherence to a comparable therapy plan (that is, 225 minutes per week) using different (e.g. non-gamified) approaches? Given that the theoretical advantage of gamified therapy is that it should be more appealing and thus increase adherence, this is worrying. This should also perhaps be described in the discussion in more detail, comparing it with related approaches and potential solutions.

Thank you for raising this interesting observation and is indeed one the team has discussed and in response we have adjusted the text accordingly to:

'On average, participants trained with the Neurofenix platform four days per week, for 149 minutes per week. Twenty-seven per cent achieved at least 225 minutes training per week which equates to the recommended 45 minutes of training per day for five days per week. [36] It may seem surprising, given the theoretical appeal of gamified therapy and its potential advantages over traditional home exercise programmes, that a larger proportion of participants were unable to achieve the recommended minimal dose. However, as community-based rehabilitation opportunities for stroke survivors typically reduce over time and rarely continue beyond six months [37] devices such as the Neurofenix platform offer a tangible means of increasing upper limb training hours from what is likely to be a very low base. [38] We do not have community based therapy comparison figures for other forms of UL therapy, but in the acute setting, where arguably patients have more access to therapists and UL treatment, only 7.9 minutes a day of combined occupational therapy and physiotherapy for the UL during the first few weeks post stroke has been reported.[39] Our participants also achieved a median (minimum-maximum) of 15,092 (222-43,999) UL movements over 7-weeks, which compares favourably to an average of 32 movement repetitions per session of inpatient upper limb rehabilitation reported by Lang et al.[40] So while adherence is variable in this study, problems related to persistence with technology-enhanced upper limb rehabilitation are consistent with previous studies using rehabilitation technology [41,42]. A recent systematic review concluded that further research is necessary to understand factors underlying perseverance with home-based technology-enhanced upper limb training post-stroke.[43] Standen et al [42] reported that family support, where available, was crucial to successful game-play. We also found that participants with severe UL impairment were dependent on carers for set-up. Participants starting from a low baseline may apply an equivalent effort as those with less impairment without meeting training targets which suggests that personalised training goals may also be useful.'

Reviewer: 2

This is a very clearly written manuscript and I commend the authors for their clear language and easy layout of information. I have minor suggestions only.

Thank you for your kind comment on the paper.

5. Abstract: Results do not contain the UL impairment/ activity outcomes - it would be good to read these in the abstract

Thank you. We have tried to address this good suggestion within the limits of the word count for the abstract. We have stated in the Outcomes that we collected impairment, activity and participation outcomes and in the Results we have named the Fugly-Meer and Motor Activity Log and Range of Motion.

6. Methods

Outcomes - it would be good to explain what is meant by 'a utility value using the cross-walk function' as this is not as familiar as other outcome measures

We agree this is a less well-known measure from the world of health economics. We have added more detail as follows:

'A utility value [i.e. a score] was calculated as recommended using the cross-walk function which maps the EQ-5D-3L and newer EQ-5D-5L questionnaires] ;[27] a higher value indicates better quality of life.'

7. Safety - in this description, it is not clear if pain and fatigue are adverse events. If fatigue is being called an adverse event it would be good to understand if the intervention was related to increased fatigue and the severity of the fatigue. It is also arguable as to whether fatigue in the context of exercise is an adverse event, as exercise that does not induce some fatigue is probably not intensive enough to be of benefit.

Thank you for highlighting these areas for clarification. We have specified that fatigue is post-stroke fatigue and have adjusted the wording throughout to make this clearer.